# Exploring multimorbidity clusters in relation to healthcare use and its impact on self-rated health among older people in India

**Salmaan Ansari**[1], **Abhishek Anand**[2], **Babul Hossain**[2]*

1 Centre for Health Services Studies, University of Kent, Kent, England, United Kingdom, 2 Department of Family and Generations, International Institute for Population Sciences, Mumbai, India

* bhossain399@gmail.com

## Abstract

The conventional definition of multimorbidity may not address the complex treatment needs resulting from interactions between multiple conditions, impacting self-rated health (SRH). In India, there is limited research on healthcare use and SRH considering diverse disease combinations in individuals with multimorbidity. This study aims to identify multimorbidity clusters related to healthcare use and determine if it improves the self-rated health of individuals in different clusters. This study extracted information from cross-sectional data of the first wave of the Longitudinal Ageing Study in India (LASI), conducted in 2017–18. The study participants were 31,373 people aged $\geq$ 60 years. A total of nineteen chronic diseases were incorporated to identify the multimorbidity clusters using latent class analysis (LCA) in the study. Multivariable logistic regression was used to examine the association between identified clusters and healthcare use. A propensity score matching (PSM) analysis was utilised to further examine the health benefit (i.e., SRH) of using healthcare in each identified cluster. LCA analysis identified five different multimorbidity clusters: *relatively healthy'* (68.72%), *'metabolic disorder* (16.26%), *'hypertension-gastrointestinal-musculoskeletal'* (9.02%), *'hypertension-gastrointestinal'* (4.07%), *'complex multimorbidity'* (1.92%). Older people belonging to the *complex multimorbidity* [aOR:7.03, 95% CI: 3.54–13.96] and *hypertension-gastrointestinal-musculoskeletal* [aOR:3.27, 95% CI: 2.74–3.91] clusters were more likely to use healthcare. Using the nearest neighbor matching method, results from PSM analysis demonstrated that healthcare use was significantly associated with a decline in SRH across all multimorbidity clusters. Findings from this study highlight the importance of understanding multimorbidity clusters and their implications for healthcare utilization and patient well-being. Our findings support the creation of clinical practice guidelines (CPGs) focusing on a patient-centric approach to optimize multimorbidity management in older people. Additionally, finding suggest the urgency of inclusion of counseling and therapies for addressing well-being when treating patients with multimorbidity.

**Data Availability Statement:** The datasets generated and/or analyzed during the current study are available in the International Institute for Population Sciences (IIPS), Mumbai repository. A

reasonable request for data access can be made at https://www.iipsindia.ac.in/content/LASI-data.

**Funding:** The authors received no specific funding for this work.

**Competing interests:** The authors have declared that no competing interests exist.

## Introduction

Multimorbidity (i.e., the co-existence of two or more chronic conditions in an individual) is becoming progressively more common with advancing age and poses a key challenge to healthcare systems worldwide [1]. Given that the proportion of older people is expected to increase from 8% in 2015 to 19% in 2050, there may be a dramatic increase in the risk of developing multiple chronic conditions, such as cardiovascular diseases, diabetes, respiratory disorders, and musculoskeletal conditions over the coming decades [2]. When these conditions coexist in an individual, they often interact synergistically, resulting in poorer health outcomes and increased healthcare needs [3–7]. Moreover, individuals with multiple chronic conditions in India experienced higher rates of hospitalization, polypharmacy, outpatient visits and higher out-of-pocket expenditure (OOPE), and medical regimes [8–12]. These studies have shed light on the healthcare challenges faced by individuals with multimorbidity but have often relied on counting the number of concurrent conditions or examining the coexistence of two or more conditions. This approach may not fully capture the complexity of the patient's healthcare needs and clinical management [13]. It is widely recognized that multimorbidity is highly heterogeneous and patients can experience a wide array of different combinations of diseases [14]. Therefore, the conventional definition of multimorbidity which typically requires the presence of two or more chronic conditions may not adequately capture the intricate treatment requirements necessitated by the complex interactions between multiple conditions [13].

In India, some studies have delved into the specific combinations of chronic non-communicable diseases (NCDs) and they used simply dyad or triad approach, or created dummy variables for identifying the pairs of the diseases [15–17]. While these methods are relatively simple and straightforward, they are typically limited to studying specific relationships, such as co-occurrence, and may not capture the complex spectrum of multimorbidity. Specifically, this approach may overlooked the broader context and potential synergistic effects of multiple health conditions on healthcare utilisation. In this context, there are some evidences from high-income countries utilised more complex statistical techniques such as Latent Class Analysis (LCA), network analysis, and k-means clustering and exploratory factor analysis [18–20]. These studies identified clinically justifiable multimorbidity patterns and observed variation in associated outcomes and healthcare needs across these clusters, providing insights into how certain conditions cluster together and their impact. These methods are often regarded as capable of uncovering complex and multivariate multimorbidity patterns. To our knowledge, no similar study has been conducted in India, with the exception of study by Puri et al. that was limited to examining patterns and associated factors among people aged 45 years or above [17]. From an etiological and clinical perspective, there is still limited knowledge to understand what makes health conditions tend to co-occur among older population, particularly in countries experiencing a growing aging population like India. Additionally, there is a need to comprehend the healthcare needs associated with specific cluster of conditions in this population.

In addition, one question still needs to be addressed; how does healthcare use impact individuals across different multimorbidity clusters? The question is also relevant, given the significance of self-rated health as a predictor of mortality, functional health and the use of healthcare services among older population. Self-rated health following healthcare use may indeed an important aspect to provide valuable information on the outcomes and effectiveness of the healthcare services provided to individual with multimorbid conditions. In this context, self-report health may hinder the complexity of their health conditions and understanding the required level of care across the multimorbidity clusters [21, 22]. Although, evidence suggests that individuals with greater numbers of chronic conditions reported lower self-rated health, highlighting the impact of multimorbidity on overall well-being [23, 24]. Assessing the impact

of healthcare use on self-reported health across the multimorbidity cluster may enable a comprehensive understanding of how healthcare use affects individuals with different multimorbidity clusters. This is also important as this perspective provides valuable insight into the unique healthcare requirements of each cluster, ultimately contributing to the improvement of health outcomes and well-being, particularly among the elderly.

In light of these available pieces of evidence, there is a knowledge gap to understand the epidemiology of multimorbidity clusters that extends beyond simply counting diseases. This gap hinders the development of effective interventions to address the complex healthcare needs of the aging population in India, particularly for those living with multimorbidity. To fill this important knowledge gap in India, the current study aimed to explore multimorbidity clusters based on patterns of disease co-occurrence concerning healthcare use and self-rated health. The conceptual framework of this study was depicted in **Fig 1**. The framework also employs covariates for two purposes: firstly, for controlling potential confounding factors, and secondly, for estimating propensity scores to address selection biases. Based on the rationale outlined above, the following research questions were addressed in this study:

1. What are the distinct clusters of chronic diseases co-occurrence among Indian older population.

2. How do different multimorbidity clusters influence healthcare use among older population?

3. How does healthcare use impact self-rated health among older individuals with different multimorbidity clusters?

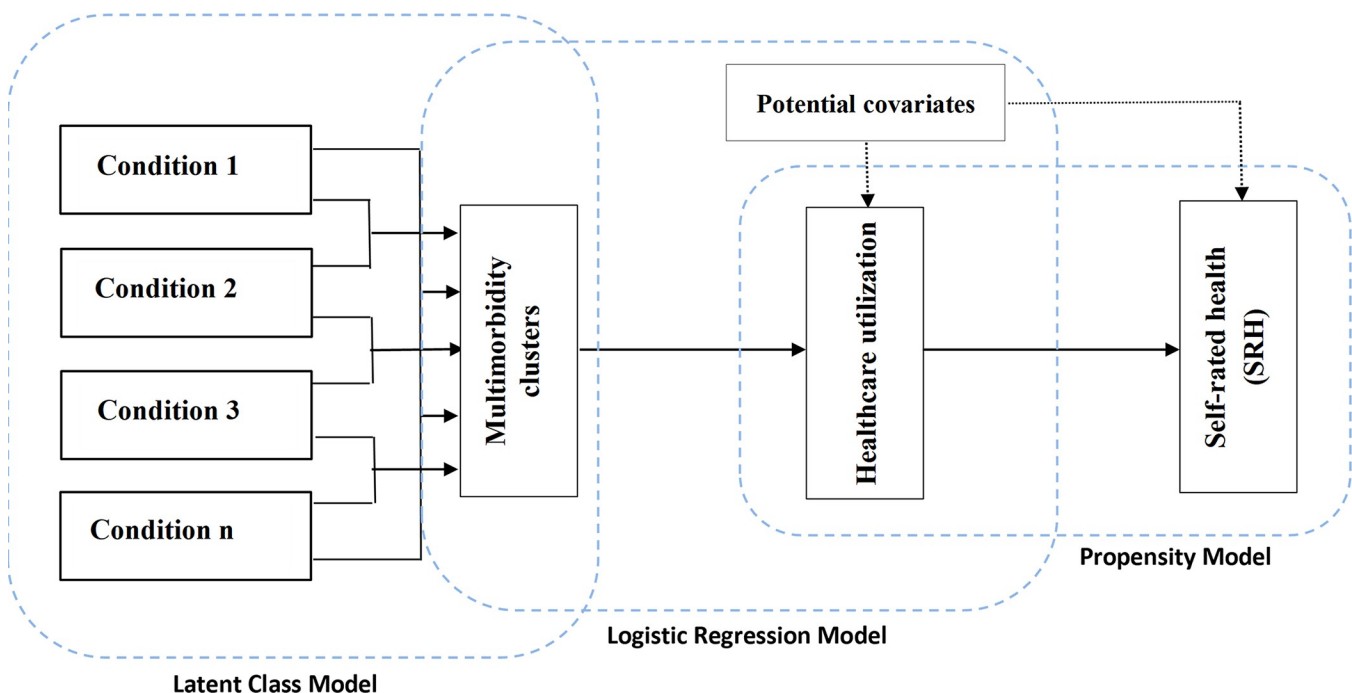

**Fig 1. Conceptual framework of the study.**

## Material & methods

### Data source and study sample

This study was based on cross-sectional data from the first wave of the Longitudinal Aging Study in India conducted in 2017–18. LASI is a large nationally representative survey covering people aged 45 and above in India and part of a global family of longitudinal health and aging studies in more than 30 countries. The prime objective of the LASI survey is to provide information on the health, economic, social, and psychological behaviors of older adults in India and its states and union territories. In the first wave, LASI adopted a multistage stratified area probability cluster sampling design. A total participant of 72,250 individuals aged ≥45 years and their spouses (irrespective of age) from all states (except Sikkim) and Union Territories (UTs) were included in the survey. Separately written informed consent was obtained for household and individual surveys and dried blood spot collection. The ethical guidelines for data collection in LASI were approved by all collaborating institutions and the Indian Council of Medical Research (ICMR). Detailed information on ethical protocols, survey design, and sampling procedure data collection is available in the LASI India report [25]. In the present study, we included 31,464 people aged 60 years or more.

### Measures

#### Healthcare use

Information regarding the healthcare use from any medical facilities or medical providers was collected based on the following question: "*In the past 12 months, have you visited any healthcare facility or any health professional visited you*?". Healthcare providers were public facilities (e.g., Primary, Urban, Community health centers, District, Government/tertiary hospitals, etc.,) and private facilities (e.g., Private hospital/nursing home, non-governmental organization (NGO)/Charity/Trust/Church-run hospital, etc). Based on the above responses, we constructed a binary variable for healthcare use and were coded as 1 "Yes" if respondents have visited any of the above healthcare facilities and 0 "No" if they have not visited any healthcare facilities.

#### Chronic health conditions

A total of 19 chronic diseases were included as indicators of multimorbidity in the study population from responses to the question: "*Has any health professional ever diagnosed you with the following chronic conditions or diseases*?". There is a list of nineteen self-reported chronic diseases included in this study. These are hypertension, chronic heart diseases (CHD), chronic obstructive pulmonary disease (COPD), stroke, chronic bronchitis, asthma, diabetes, cancer or malignant tumor, arthritis, rheumatism, osteoporosis, mental disorders, thyroid disease, urinary incontinence, gastrointestinal disorders, skin disease, chronic renal failure, kidney stones, and high cholesterol. Based on presence or absence, the above-diagnosed diseases were coded as 1 "Yes" and 0 "No" and further a list of chronic conditions was used in Latent Class Analysis to be grouped into clusters. Clusters with two or more conditions were defined as multimorbidity clusters [26].

#### Self-rated health (SRH)

In epidemiological research, self-rating of health is among the most frequently assessed health perceptions and has been considered the strongest predictor of future morbidity and mortality [27]. Therefore, SRH was defined by using a single question "*Overall, how is your health in general*?" with five response categories such as 1 "Very poor", 2 "Poor", 3 "Fair", 4 "Good", and 5 "Very Good". The higher score indicates a higher level of SRH of an individual's perceived health.

**Table 1. Description of the covariates from the LASI survey, 2017–18.**

| Covariates | Category |
| --- | --- |
| **Place of residence** | Rural<br>Urban |
| **Age group** | 60-69 years<br>70-79 years<br>80 years or more |
| **Wealth status**[1] | Poorest<br>Poor<br>Middle<br>Richer<br>Richest |
| **Religion** | Hindu<br>Muslim<br>Other (Sikh, Christian, and others) |
| **Social groups**[2] | Scheduled caste (SC)<br>Scheduled tribe (ST)<br>Other backward class (OBC)<br>Other |
| **Gender** | Male<br>Female |
| **Education** | No education<br>Less than 5 years<br>5-9 years<br>10 or more years |
| **Marital status** | In union<br>Not in union |
| **Living alone** | No<br>Yes |
| **Region** | North<br>Central<br>East<br>North East<br>West<br>South |

[1]Information on the wealth status of the respondents was determined using the monthly per capita expenditure (MPCE) of the households

[2]Social group like SC, ST & OBC are legally designated groupings of individuals among India's most disadvantaged socioeconomic groups experiencing stigma, limited access to education, lower asset holding, and reduced access to health and healthcare.

## Covariates

Several potential covariates were selected from the survey including demographic and socio-economic factors. The covariates used for this study from the LASI survey are described in **Table 1**.

## Statistical analysis

Of the 31,464 individuals aged 60 years or more who responded to the survey, 91 had missing data on at least one study variable. After excluding missing data from the database, resulting in a final sample of 31,373 individuals aged 60 years or more (**Fig 2**). Initially, descriptive statistics of the study variables and chronic health conditions were presented.

**Identification of multimorbidity clusters (Latent class model).** Furthermore, Latent class analysis (LCA) was utilized to identify multimorbidity clusters. The main purpose of this

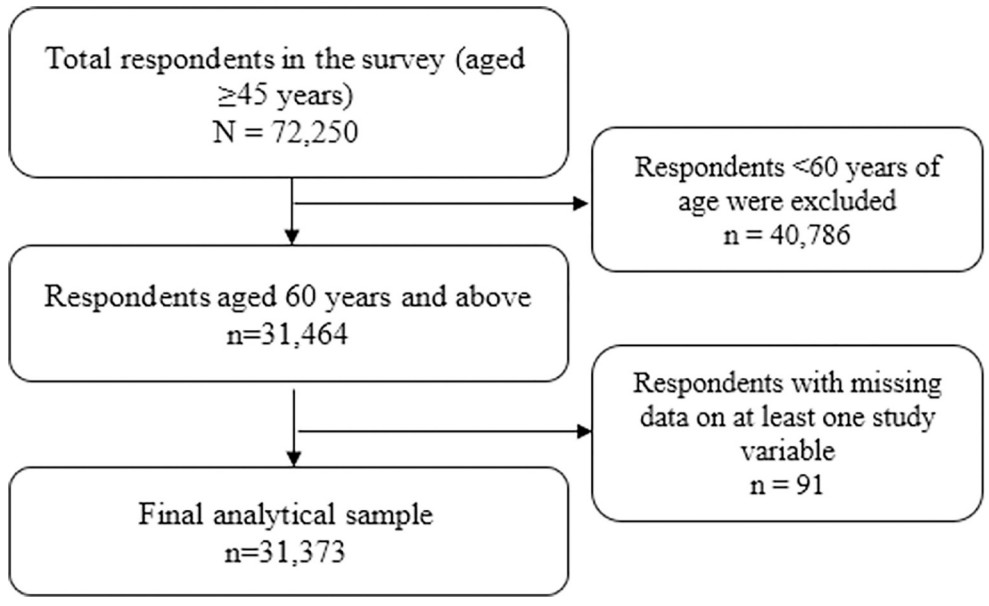

**Fig 2. Flow chart of the sample selection process.**

analysis was to determine whether healthcare use varies across different multimorbidity clusters. Adjusted Bayesian Information Criteria (BIC) and the consistent Akaike Information Criteria (CAIC) were considered to determine the optimal number of latent classes [28]. The likelihood-ratio G2 statistic (and parametric bootstrap likelihood ratio test) were used to test the null hypothesis that the specified LCA model fits the data (i.e., a significant p-value indicated that the null model was too restrictive) [29]. The adjusted BIC and CAIC are more robust indicators of class enumeration with categorical outcomes and were used to compare several plausible models where the lowest values indicate the best-fitting model [30]. After the selection of an optimal model, interpretability and clinical judgment were required, therefore, item-response probabilities (i.e., the estimated probability of reporting a particular NCD, given membership in particular latent class) were utilized to assign labels to the identified latent disease classes, i.e., the labels were based on the item(s) (disease(s)) with higher probabilities [18, 31]. Item-response probabilities of 30% or higher were deemed to be the optimal fit following suggestions from existing research [32]. Therefore, each respondent was assigned to their best-fit class based on the maximum item-response probability of each latent class calculated as an indicator of classification certainty. Finally, an independent variable containing identified multimorbidity clusters was created.

**Examination of the associations between multimorbidity clusters and healthcare use (logistic regression model).** Binary logistics regression models were fitted to examine univariate (odds ratios [ORs]) and adjusted associations (adjusted odds ratios [aORs]) between identified multimorbidity clusters and the outcome of healthcare use. We also estimated 95% confidence intervals (95% CI). The equation of the logistic regression, considering healthcare use as the outcome variable ($y$) were as follows:

Crude model:

$$logit(y) = \ln\left(\frac{p}{1-p}\right) = \beta_0 + \beta_1 X_1$$

Adjusted model:

$$logit(y) = \ln\left(\frac{p}{1-p}\right) = \beta_0 + \beta_1 X_1 + \beta_2 X_2 + \beta_3 X_3 + \cdots + \beta_k X_k$$

The crude model consists of the intercept term ($\beta_0$) and the regression coefficient ($\beta_1$) for the primary independent variable of interest which was the multimorbidity cluster ($X_1$). The adjusted model extends the crude model by including additional independent variables ($X_2$, $X_3$,. . ...$X_k$) as covariates that are sociodemographic variables. Each covariate has its regression coefficient ($\beta_2$, $\beta_3$,. . ., $\beta_k$). This allows for examining the adjusted associations between healthcare use and the multimorbidity cluster while controlling for the effects of sociodemographic variables included as covariates.

**Investigation of the impact of healthcare use on the level of SRH within each multimorbidity cluster (propensity model).** In this section of the study, we focused on five different subgroups based on identified multimorbidity clusters. For each cluster, we employed the propensity score matching (PSM) method to investigate how healthcare utilization affects the level of SRH among older people. Healthcare use was considered as a treatment variable and SRH as an outcome variable. Firstly, a logistic regression model stratified by each subsample (cluster) was used to estimate all participants' propensity scores for healthcare use based on the selected study covariates. Secondly, estimated propensity scores were used to match the study subsample using the nearest neighbours matching algorithm. It can evaluate the balance after propensity score matching in which the treatment group is 'participants using healthcare' and the control is 'participants not using healthcare. The formula is given below:

$$D_i = Y_{1i} - Y_{0i}$$

Where $D_i$ is the difference between outcome (SRH) for an $i^{th}$ individual with treatment (healthcare use) and without treatment (no healthcare use) and ($Y_{1i}$, $Y_{0i}$) demonstrates the SRH if an $i^{th}$ individual uses healthcare or does not use healthcare. It is also known as the treatment effect (causal effect of the treatment). Alternatively, we estimated the average treatment effect (ATE) and average treatment effect on treated (ATET). The formula for these two estimates follows:

$$ATE = E(D_i) = E(Y_1 - Y_0) = E(Y_1) - E(Y_0)$$

Where E ($Y_1$) is the expected value of $Y$ for all the units in the treatment group and E($Y_0$) is the expected value of $Y$ for all the units in the control group.

$$ATET = E(Y_1 - Y_0 | T = 1) = E(Y_1 | T = 1) - E(Y_0 | T = 0)$$

In the above equation, $T = (0,1)$ refers to the control and treatment groups. *ATET* is the difference in average actual outcomes between treatment and control groups. However, $E(Y_0 | T = 1)$ refers to a contrafactual mean for those being not treated is not observed.

Sample survey weights were used in the analysis. Statistical analysis was done in Stata version 17.0 and the significance level was set at a two-sided p-value of 0.05 [33].

## Ethics approval

The study was performed as per the Helsinki Declaration and the national and international guidelines. The necessary guidance and ethical guidelines in the LASI survey were approved by the Indian Council of Medical Research (ICMR), India. More details on the guidelines and protocols are available in the LASI India report.

## Results

### Sample characteristics and morbidity conditions among older people in India

Nearly two-thirds of the respondents were from rural areas; almost 82% were Hindus and 12% were Muslims. In the study sample, more than half of the respondents are older women (52.80%). Almost 57% of respondents had no education whereas nearly 22% had more than ten years of education. Nearly 38% of respondents were not in a union and about 6% were living alone (see **Table 2**). The most common chronic conditions were hypertension (32.78%), gastrointestinal disorders (19.10%), and diabetes (14.25%). Other common chronic conditions were arthritis (10.97%), and rheumatism (7.86%), skin diseases (5.24%) (see **Fig 3**).

### Multimorbidity clusters among older people in India

**Fig 4** shows the value of adjusted BIC and CAIC for the one to five-class LCA models for older people. There was a considerable decline in the adjusted BIC and CAIC values from the two-class to the five-class model and the above five-class models were not well-identified indicating the five-class model is the optimum. **Fig 5** presents the item-response probabilities for the five-class model for older people. Based on the item-response probabilities, we found five multimorbid clusters and they have been assigned descriptive labels to characterize their comorbid characteristics. Age and sex-adjusted prevalence of each cluster were given in **Fig 6**. *Class 1* was labelled as "*relatively healthy*" as it was characterized by individuals with low probabilities of all 19 chronic conditions and the majority of the study sample (68.37%) were classified as being in this "relatively healthy" class. *Class 2* was labelled as a '*metabolic disorder*' as it was classified by individuals with a high probability of hypertension and diabetes and it comprised 16.48% of study sample. *Class 3* was labelled as '*hypertension/gastrointestinal/musculoskeletal*' as it included those with a high probability of hypertension, and gastrointestinal, and musculoskeletal disorders and comprised 9.04% of the study sample. *Class 4* was labelled as '*hypertension/musculoskeletal*' as it was characterized by an increased probability of hypertension and musculoskeletal disorder. A remaining pattern which is in *Class 5* was labelled as '*complex multimorbidity*' as it included ≥3 health conditions and comprised 1.94% of study sample.

### Association between multimorbidity clusters and healthcare use among older people

**Table 3** illustrates the to examine the association between identified multimorbidity clusters and healthcare use among older people. Older people with '*Complex multimorbidity*' cluster (94.35%) reported higher healthcare use, followed by '*hypertension/gastrointestinal*' cluster (89.69%), '*hypertension/gastrointestinal/musculoskeletal*' cluster (89.51%) and '*metabolic disorders*' cluster (87.06%). Results from binary logistic regression analysis showed that compared with the '*relatively healthy*' cluster, healthcare use was significantly higher for each multimorbidity cluster even after controlling potential covariates. The odds ratio for the "*complex multimorbidity*" cluster in both the crude and adjusted model was relatively higher ([OR:7.05, 95% CI: 3.81–13.02] and [aOR:7.03, 95% CI: 3.54–13.96], respectively) followed by '*hypertension/gastrointestinal/musculoskeletal*' cluster ([OR:3.60, 95% CI: 3.03–4.29] and [aOR:3.27, 95% CI: 2.74–3.91], respectively), '*hypertension/gastrointestinal*' cluster ([OR:3.67, 95% CI: 2.90–4.66] and [aOR:3.01, 95% CI: 2.35–3.82], respectively) and '*metabolic disorders*' cluster ([OR: 2.84, 95% CI: 2.33–3.46] and [aOR: 2.78, 95% CI: 2.29–3.38], respectively).

**Table 2. Socio-demographic profile of the study participants for older people in India, 2017–18.**

| Sample Characteristics | N | % |
|---|---|---|
| **Place of residence** | | |
| Rural | 20,682 | 70.87 |
| Urban | 10,691 | 29.13 |
| **Age group** | | |
| 60–69 years | 18,926 | 58.73 |
| 70–79 years | 9,072 | 29.94 |
| 80 years or more | 3,375 | 11.33 |
| **Wealth status**[*] | | |
| Poorest | 6,462 | 21.79 |
| Poor | 6,456 | 21.78 |
| Middle | 6,397 | 20.63 |
| Richer | 6,157 | 19.29 |
| Richest | 5,901 | 16.51 |
| **Religion** | | |
| Hindu | 23,040 | 82.66 |
| Muslim | 3,720 | 10.94 |
| Other | 4,612 | 6.41 |
| **Social groups** | | |
| SC | 5,127 | 18.98 |
| ST | 5,159 | 8.15 |
| OBC | 11,850 | 45.02 |
| Others | 9,237 | 27.86 |
| **Gender** | | |
| Male | 15,036 | 47.20 |
| Female | 16,337 | 52.80 |
| **Education** | | |
| No Education | 16,853 | 56.76 |
| Less than 5 years | 3,772 | 11.48 |
| 5–9 Years | 5,997 | 17.89 |
| 10 or more years | 4,751 | 13.86 |
| **Marital status** | | |
| In Union | 20,032 | 61.92 |
| Not in Union | 11,341 | 38.08 |
| **Living alone** | | |
| No | 29,756 | 94.39 |
| Yes | 1,617 | 5.71 |
| **Region** | | |
| North | 5,799 | 12.65 |
| Central | 4,259 | 21.07 |
| East | 5,749 | 23.76 |
| Northeast | 3,739 | 2.97 |
| West | 4,282 | 17.21 |
| South | 7,545 | 22.34 |
| **Total** | **31,373** | **100.0** |

SC: Scheduled caste; ST: Scheduled tribe; OBC: Other backward class

N: Frequency; %: Percentages; Frequencies are unweighted; Percentages are weighted.

* Wealth status was based on monthly per capita expenditure (MPCE)

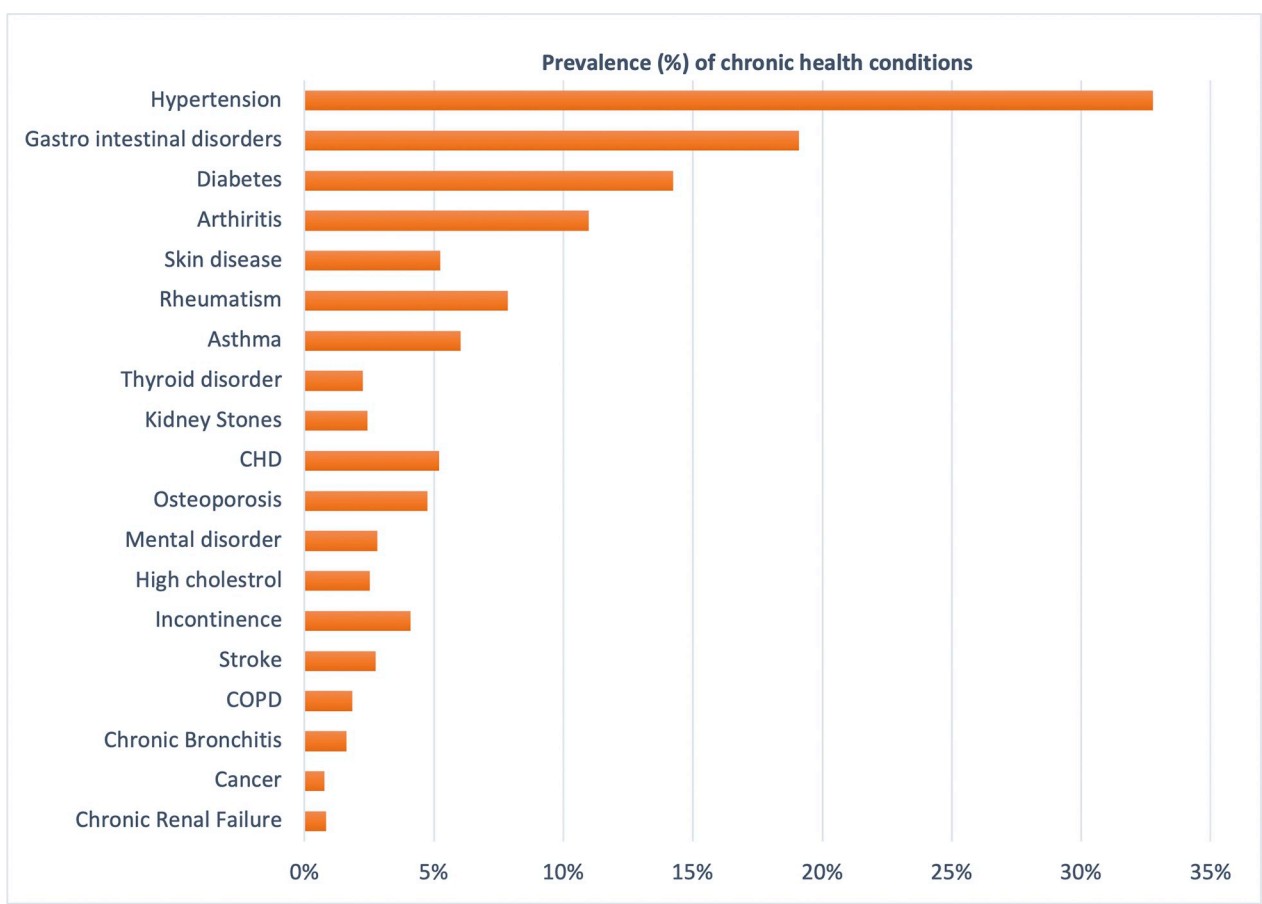

**Fig 3. Prevalence of chronic health condition among older people in India, 2017–18.**

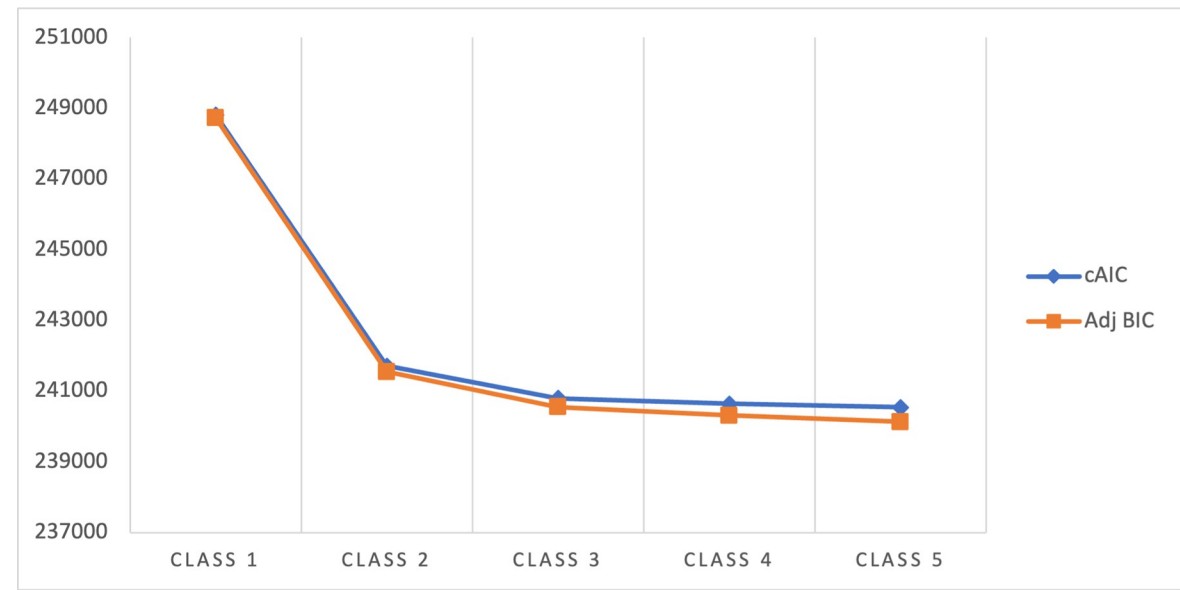

**Fig 4. Relative fit for latent class analysis (cAIC, adjusted BIC).**

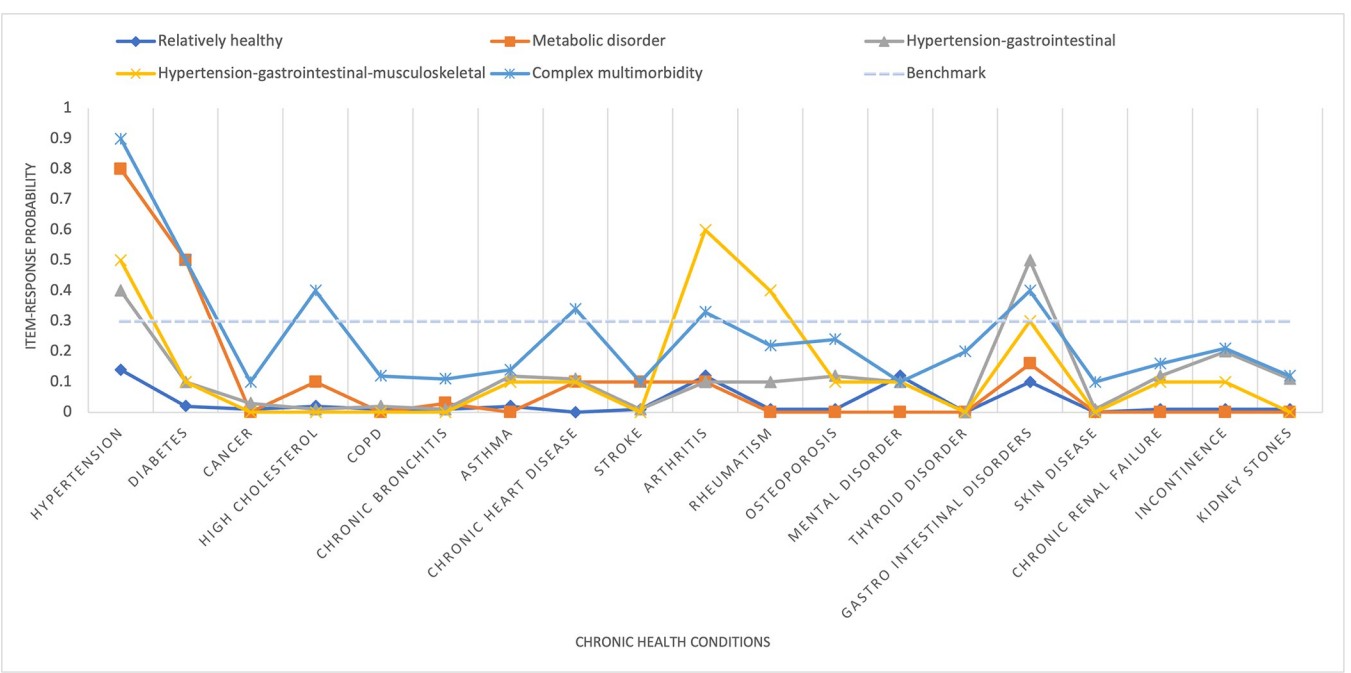

**Fig 5. Probabilities of having chronic conditions for each latent class for older people in India, 2017–18.**

## Impact of health care use on the level of SRH within each multimorbidity cluster among older people in India

First, the estimated ATT of healthcare use on SRH for each cluster was presented in **Table 4**. Using the nearest neighbor matching method demonstrated that healthcare use was significantly associated with a decline in SRH in each multimorbidity cluster. In the '*complex multimorbidity*' cluster, there is a significant decline in the level of SRH (i.e., ATT = -0.4324, T-value = -1.94) followed by the '*hypertension/gastrointestinal*' cluster (ATT = -0.4148, T-value = -3.36), '*hypertension/gastrointestinal/musculoskeletal*' cluster (ATT = -0.3778, T-value = -5.14), and '*metabolic disorders*' cluster (ATT = -0.3553, T-value = -6.64). Alternatively, similar results in **Table 5** were shown by the coefficient of ATE and ATET which means that across multimorbidity clusters, even after using healthcare, the level of SRH is declined. For the '*complex multimorbidity*' clusters, the value of ATE and ATET was -0.5643 (p-value<0.05) and -0.5954 (p-value<0.05), respectively, signified that healthcare use decreased the level of SRH by 56.43% and the propensity score for a high level of SRH was 59.54% points lower among those used healthcare.

## Discussion

The present study examined multimorbidity clusters in older people, healthcare utilization across these clusters, and the impact of healthcare use on their self-rated health. A nationally representative dataset of 31,373 individuals aged 60 years or more was thoroughly analyzed, incorporating information on 19 chronic conditions. The study identified five distinct multimorbidity clusters with prevalence rates adjusted for age and sex: '*relatively healthy*' (68.72%), '*metabolic disorder*' (16.26%), '*hypertension-gastrointestinal-musculoskeletal*' (9.02%), '*hypertension-gastrointestinal*' (4.07%), and '*complex multimorbidity*' (1.92%). Additionally, our study observed variations in the strength of the association between multimorbidity clusters

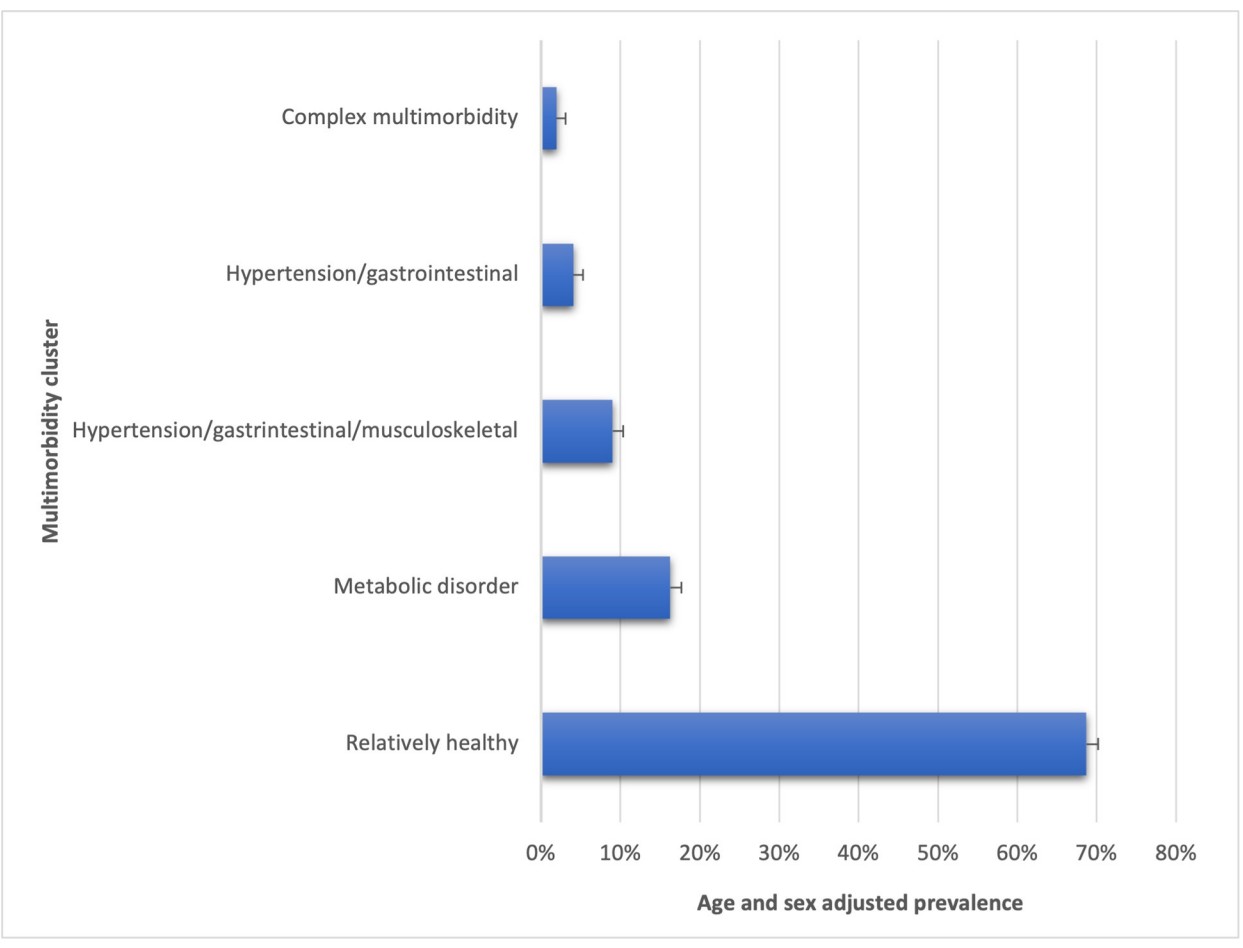

**Fig 6. Age and sex adjusted prevalence od identified multimorbid clusters among older people in India, 2017–18.**

and healthcare use in older people, and the association was stronger for people with cluster *hypertension-gastrointestinal-musculoskeletal* and *complex multimorbidity*. Surprisingly, results from the PSM analysis indicated that healthcare use did not lead to an improvement in perceived health status among older individuals with identified multimorbidity clusters.

**Table 3. Association between multimorbidity clusters and healthcare use for older people in India, 2017–18.**

| Latent class | Multimorbidity cluster | Healthcare use (%)* | Crude model | Adjusted model |
|---|---|---|---|---|
| | | | OR [95% CI] | OR [95% CI] |
| *Class 1* | *Relatively Healthy (Ref.)* | 70.31 | 1 | 1 |
| *Class 2* | *Metabolic disorder* | 87.06 | 2.84* [2.33–3.46] | 2.78* [2.29–3.38] |
| *Class 3* | *Hypertension/Gastrointestinal/musculoskeletal* | 89.51 | 3.60* [3.03–4.29] | 3.27* [2.74–3.91] |
| *Class 4* | *Hypertension/Gastrointestinal* | 89.69 | 3.67* [2.90–4.66] | 3.01* [2.35–3.82] |
| *Class 5* | *Complex multimorbidity* | 94.35 | 7.05* [3.81–13.02] | 7.03* [3.54–13.96] |

* Significant using chi-square test; OR: odds ratio estimated from logistic regression; 95% CI: 95% Confidence intervals; Ref: Reference category; *p<0.05

All models were adjusted for sociodemographic variables including gender, education, marital status, place of residence, wealth status religion, caste, and region.

**Table 4. Results from PSM for the role of healthcare use on the SRH in subsamples stratified by multimorbidity clusters for older people in India, 2017–18.**

| Latent class | Subsamples stratified by Multimorbidity clusters | Treatment | Outcome: SRH | | | | |
|---|---|---|---|---|---|---|---|
| | | | Treated | Control | Difference | S.E. | T-value |
| Class 1 | Relatively Healthy | Healthcare use | 2.61 | 2.99 | -0.38 | 0.02 | -23.84* |
| Class 2 | Metabolic disorder | Healthcare use | 2.34 | 2.70 | -0.36 | 0.05 | -6.64* |
| Class 3 | Hypertension/Gastrointestinal/musculoskeletal | Healthcare use | 2.24 | 2.62 | -0.38 | 0.07 | -5.14* |
| Class 4 | Hypertension/Gastrointestinal | Healthcare use | 2.10 | 2.52 | -0.41 | 0.12 | -3.36* |
| Class 5 | Complex multimorbidity | Healthcare use | 1.81 | 2.24 | -0.43 | 0.22 | -1.94* |

Note-Each latent class or cluster is presented as a subsample; S.E: Standard error

*p<0.05

Among the identified clusters, the *relatively healthy* cluster is consistent with the previous studies reporting a similar latent class representing the majority of the study sample, ranging from 50 to 70 percent [17, 34–37]. The second most prevalent cluster, the *metabolic disorders* cluster, comprising comorbidity of hypertension and diabetes, is also supported by previous studies [17, 37]. We speculated that this cluster may be caused by increased risk factors for both diabetes and hypertension such as obesity (or an unbalanced diet rich in fat and salt, for example), excessive carbohydrate and alcohol intake prevalent in the Indian population [38–40]. Moreover, there has been a well-described association between diabetes and hypertension by sharing common metabolic abnormalities and pathophysiology [41, 42]. The third identified cluster *hypertension-gastrointestinal* may be due to the impact of metabolism-related risk factors on the gastrointestinal tract, a crucial organ in metabolic syndrome and metabolic hypertension (i.e., hypertension due to metabolic disturbance Studies have indicated that the gastrointestinal tract plays a significant role in both metabolic syndrome and hypertension, explaining the clustering of gastrointestinal disorders and hypertension among older individuals in India [43, 44]. Furthermore, medications used for musculoskeletal disorders, such as non-steroidal anti-inflammatory drugs for arthritis, can have a strong adverse effect on gastrointestinal-related organs like the stomach [45]. This could increase the risk of stomach or digestive disorders in individuals with musculoskeletal conditions, potentially contributing to the identification of the fourth cluster of *hypertension-gastrointestinal-musculoskeletal* in the current study. Noteworthy, hypertension stood out as the prevalent health condition among those with existing chronic conditions, suggesting that individuals diagnosed with hypertension could be grouped with other chronic diseases, heightening the risk of health complications.

**Table 5. Estimated ATE and ATET of healthcare use on the level of SRH across identified multimorbidity clusters for older people in India, 2017–18.**

| Latent class | Subsamples stratified by Multimorbidity clusters | ATE | | ATET | |
|---|---|---|---|---|---|
| | | Coefficient [95% CI] | Robust S.E. | Coefficient [95% CI] | Robust S.E. |
| Class 1 | Relatively Healthy | -0.37* [-0.41 to -0.34] | 0.02 | -0.38* [-0.42 to -0.35] | 0.02 |
| Class 2 | Metabolic disorder | -0.37* [-0.46 to -0.29] | 0.05 | -0.37* [-0.47 to -0.28] | 0.05 |
| Class 3 | Hypertension/Gastrointestinal/musculoskeletal | -0.48* [-0.61 to -0.36] | 0.06 | -0.48* [-0.61 to -0.35] | 0.07 |
| Class 4 | Hypertension/Gastrointestinal | -0.40* [-0.77 to -0.04] | 0.18 | -0.41* [-0.78 to -0.03] | 0.19 |
| Class 5 | Complex multimorbidity | -0.56* [-0.78 to -0.35] | 0.11 | -0.60* [-0.82 to -0.37] | 0.11 |

ATE: Average treatment effect; ATET: Average treatment effect on the treated; 95% CI: 95% Confidence intervals; S.E.: Standard error

*p<0.05.

Limited studies explored healthcare use in different combinations of chronic diseases, reporting variations across multimorbidity clusters, but no such study exists for India [18–20]. Our research aligns with this trend and further reveals that the identified multimorbidity clusters differ in the extent of their healthcare usage. For instance, older people belonging to the *complex multimorbidity* cluster, followed by the *hypertension-gastrointestinal-musculoskeletal* cluster, were more likely to use healthcare services than individuals with *relatively healthy* cluster. It is speculated that multimorbidity clusters involving three or more conditions affecting at least three different organ systems in one person may require regular care and higher inpatient services for assessing their health status and implementing secondary prevention [46]. This is reinforced by a previous study, which reported that older people in India with three or more conditions were 1.5 times more likely to utilize healthcare, including both inpatient and outpatient care [8]. Furthermore, studies have indicated that age-related musculoskeletal disorders, such as arthritis, are associated with increased functional disability and a higher probability of seeking medical attention, undergoing outpatient surgeries, experiencing hospitalization events, and requiring more ambulatory care [47–49]. Moreover, when combined with other health conditions like hypertension and gastrointestinal disorders, the situation becomes more complex, potentially increasing the demand for medical care. This finding suggests that Indian healthcare systems and providers need to be prepared to handle the growing complexity and demand for integrated, multidisciplinary care.

Our findings also add depth to the understanding of the interplay between healthcare utilization and individuals' perception of their health within each multimorbidity cluster. The findings are also interesting in the context that, it is usually assumed that using health care not only improves an individual's physical health but also helps to improve overall well-being [50]. Contrary to this common assumption, our results indicate that higher outpatient and inpatient healthcare use with multimorbidity does not necessarily result in an improvement in their subjective perception of their health [51, 52]. It may be speculated that the association between healthcare utilization and perceived health is influenced by various factors beyond the mere utilization of healthcare services. Various aspects such as the nature and severity of chronic conditions, psychological factors, social support, and quality of care received may also play significant roles in shaping individuals' perception of their health [24, 53–56]. This finding also shows the lack of communication and dialogue between patients with the healthcare system in India. It is important to mention that, in many developed countries where therapies and community meetings are common for patients suffering certain morbidity or life-threatening diseases, there is no such concept in the Indian context [57]. This finding also shows the urgency of the inclusion of counselling and therapies for addressing well-being.

## Strength and limitations

Major strengths of the present study include the provision of a large and nationally representative sample of the older population which could be generalized to the whole country. Furthermore, this research employed a comprehensive list of nineteen chronic diseases and followed a universally accepted definition of "multimorbidity," which refers to the simultaneous presence of two or more chronic diseases. More importantly, examining possible clusters of chronic health conditions in relation to healthcare utilization and perceived health may help for improving clinical protocols and promote health and well-being. To achieve this, the research employed a widely adopted technique (LCA), which has been utilized in recent years and served as the foundation for other multimorbidity pattern studies [31, 32, 34, 36]. To do so, it applies a technique (LCA) that has been widely used in recent years and has been the basis for other studies of multimorbidity patterns.

Some limitations should be considered when interpreting the findings of the present study. Firstly, the use of self-reported measures of chronic disease and healthcare utilization may have underestimated their prevalence, especially among older individuals and those from poorer socioeconomic and educational backgrounds, who are more inclined to underreport these characteristics. Secondly, the cross-sectional nature of data limits us from inferring confident causal conclusions. Thirdly, the number of latent classes and clusters/patterns may not be comparable due to the result of variations in the characteristics of the study sample as well as the number and type of chronic diseases that were examined. Beyond the statistical technique, our investigation found some clusters that were expected and corroborated by the results of other studies carried out in India and other nations [34, 58, 59]. Fourthly, we also include participants without chronic disease or multimorbidity for comparison purposes in the context of healthcare use, which may explain the contrasting findings. Fifthly, the study lacked data on the timing of diagnoses and the severity of the conditions. Future research that includes such information in the analysis could offer valuable insights for developing more effective prevention strategies. Lastly, we could only focus on physical chronic health conditions, and the inclusion of mental health conditions in future studies may reveal different patterns and provide a more comprehensive perspective.

## Implication for practice and research

The present study indicates hypertension as a common chronic condition to be clustered with other health conditions among Indian older people. Hypertension, a manageable but often not curable, affects the cardiovascular system leading to cardiac complications and other health conditions, particularly in later life [60]. Therefore, Indian physicians should focus on lowering blood pressure by considering a comprehensive assessment of the patient's medical history and lifestyle to tailor personalized treatment plans [61]. In this context, there is a need to strengthen the implementation of high-impact and low-cost programs to diagnose and control elevated blood pressure, ensuring its comprehensive coverage to the whole country, with a particular focus on the older population [62].

Moreover, findings from this study indicate the variation in the magnitude of the association between healthcare use and different multimorbid cluster. It highlights the need for unique and a patient-centric approach in primary care settings to treat patients with multiple chronic conditions. Evidence suggested that the Indian healthcare system which is a single-disease paradigm primarily focused on acute care may not be adequate in the treatment of older people, particularly those living with multiple conditions [63]. Therefore, there is a need for more effective and patient-centric healthcare to tackle the complex interactions of chronic diseases or specific combinations of diseases that patients may accumulate. Future research should delve deeper into understanding the interactions between chronic conditions including physical and mental health, while also examining various aspects of healthcare utilization, such as inpatient and outpatient care, polypharmacy, length of hospital inpatient stays, healthcare expenditure, and medication adherence. This holistic approach will contribute to facilitate more informed decision making and multimorbidity management.

Considering subjective health following healthcare use, our study offers valuable insights into the complexity of multimorbid conditions and the effectiveness of healthcare use. This underscores the importance of urgently integrating counseling and therapies focused on addressing well-being into the treatment plans for patients with multimorbidity. A further longitudinal epidemiological study is needed to provide a more comprehensive understanding of how healthcare utilization and disease interactions impact older individuals' health and well-being in the Indian population.

## Conclusion

This study sheds light on the complexity of multimorbidity among older individuals in India, identifying five distinct clusters of chronic health conditions with differing healthcare utilization patterns. The findings underscore the significance of considering interactions between conditions, which can complicate decision-making and management strategies. Moreover, the study highlights that healthcare use is influenced by specific combinations of health conditions rather than just their quantity. Indian healthcare system which is very largely configured for acute care services and predominantly guided by single-disease pathways is not sufficient to deal with patients suffering from different combinations of diseases. Our findings support the creation of clinical practice guidelines (CPGs) focusing on a patient-centric approach to optimize multimorbidity management in older people. The noteworthy finding from the current research is that healthcare utilization did not lead to a significant improvement in the reported health statuses of older individuals. This finding shows the urgency of the inclusion of counseling and therapies for addressing well-being when treating patients with multimorbidity.

## Author Contributions

**Conceptualization:** Salmaan Ansari, Abhishek Anand, Babul Hossain.

**Data curation:** Salmaan Ansari.

**Formal analysis:** Salmaan Ansari.

**Writing – original draft:** Abhishek Anand.

**Writing – review & editing:** Salmaan Ansari, Babul Hossain.

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
