## [Decision Letter · Decision Letter 0]

10 Jul 2023

PGPH-D-23-01119

Exploring multimorbidity clusters in relation to healthcare use and its impact on self-rated health among older people in India

Dear Hossain,

Thank you for submitting your manuscript to PLOS Global Public Health. After careful consideration, we feel that it has merit but does not fully meet PLOS Global Public Health’s publication criteria as it currently stands. Therefore, we invite you to submit a revised version of the manuscript that addresses the points raised during the review process.

We look forward to receiving your revised manuscript.

Kind regards,

Collins Otieno Asweto, PhD

Academic Editor

Journal Requirements:

1. Please provide separate figure files in .tif or .eps format only and remove any figures embedded in your manuscript file. Please also ensure all files are under our size limit of 10MB.

2. We have noticed that you have uploaded Supporting Information files, but you have not included a list of legends. Please add a full list of legends for your Supporting Information files after the references list. 

Reviewers' comments:

Reviewer's Responses to Questions

**Comments to the Author**

1. Does this manuscript meet PLOS Global Public Health’s publication criteria? Is the manuscript technically sound, and do the data support the conclusions? The manuscript must describe methodologically and ethically rigorous research with conclusions that are appropriately drawn based on the data presented.

Reviewer #1: Yes

Reviewer #2: Yes

2. Has the statistical analysis been performed appropriately and rigorously?

Reviewer #1: Yes

Reviewer #2: Yes

3. Have the authors made all data underlying the findings in their manuscript fully available (please refer to the Data Availability Statement at the start of the manuscript PDF file)?

Reviewer #1: Yes

Reviewer #2: Yes

4. Is the manuscript presented in an intelligible fashion and written in standard English?

Reviewer #1: Yes

Reviewer #2: No

5. Review Comments to the Author

Reviewer #1: Reviewer Report

Topic

Very appropriate and reads fine.

Abstract

Background: This is lacking in the abstract. The authors should provide a sentence or two on the background.

Purpose/Aim: This is apt and reads well.

Methods: Appropriate.

Results: Adequate.

Conclusion: This was adequate.

Overall comment on the abstract: Except for the missing background, the abstract was adequate and sufficient.

Introduction

Background: This was well written and sufficiently addressed the topic under study. However, some minor corrections required:

Para 2, sentence 1: Source required.

Para 3, sentence 1: Source required.

Para 4, sentence 6: “Evidence also suggests reporting poor or good.” This is not very clear. It appears hanging. Either delete or recast.

Para 4, last sentence, line 3: “This is also important as this perspective of multimorbidity

clusters and healthcare use provides us with a new dimension of multimorbidity-

related healthcare requirements to further helps to improve the health status of the

elderly.” Replace “to” with “which”.

Problem statement: Clear and elucidates the problem well.

Materials and methods

Study sample: Line 9 – “Separate written informed consent was obtained for household and

individual surveys and dried blood spot collection.” Replace “separate” with

“separately”.

Chronic health conditions, page 6, sentence 2: “The list of nineteen self-reported chronic

diseases included hypertension, chronic heart diseases (CHD), chronic obstructive

pulmonary disease (COPD), stroke, chronic bronchitis, asthma, diabetes, cancer or

malignant tumor, arthritis, rheumatism, osteoporosis, mental disorders, thyroid

disease, urinary incontinence, gastrointestinal disorders, skin disease, chronic renal

failure, kidney stones, and high cholesterol.” This sentence reads well, but too

long. Recast into two sentences.

Statistical analysis: Adequate.

Results: Adequate.

Discussions

Adequate and reads well. However, some minor corrections required:

Page 12, para 2, sentence 3: “Accumulated evidence indicated that uncontrolled high blood

pressure plays a central role in the developing cardiovascular (i.e., heart attack,

stroke, diabetes, myocardial infarction, etc.) and non-cardiovascular outcomes (i.e.,

dementia, cancer, oral health conditions, etc.) [50–52].” Please, delete the article

“the”.

Page 13, para 2, sentence 1: “While, we have also provided an important finding on the

linkage between multimorbidity clustering, healthcare use, and SRH.” This sentence

does not read well. Consider a recast.

Conclusion

This is adequate and reads well.

Overall observation

On the whole, this paper was well written. The authors have demonstrated a good understanding of the subject matter and executed the study pretty well. I congratulate them for achieving this fit.

Reviewer #2: 1. The introduction section requires substantial improvement. It looks too very descriptive and not enough recent existing studies have been included. I would suggest adding more literatures and add a conceptual framework.

2. I understand that you have been given a word limit for the abstract, but I think authors should include a sentence informing the reader what the problem is and then move on to the sentence stating the aims of the paper.

3. The study does not discuss how the findings of this study connect to past research in this field. The authors should revise their Introduction to include references to the relevant literature, particularly newly published works by

Akhtar SN, Saikia N, Muhammad T (2023) Self-rated health among older adults in India: Gender specific findings from National Sample Survey. PLoS ONE 18(4): e0284321. https://doi.org/10.1371/journal.pone.0284321

Khan, M.R., Malik, M.A., Akhtar, S.N. et al. Multimorbidity and its associated risk factors among older adults in India. BMC Public Health 22, 746 (2022). https://doi.org/10.1186/s12889-022-13181-1

Akhtar, S.N. and Saikia, N. (2022), "Differentials and predictors of hospitalisation among the elderly people in India: evidence from 75th round of National Sample Survey (2017-2018)", Working with Older People, Vol. 26 No. 4, pp. 325-341. https://doi.org/10.1108/WWOP-11-2021-0055

Ahmed W, Muhammad T, Maurya C, Akhtar SN (2023) Prevalence and factors associated with undiagnosed and uncontrolled heart disease: A study based on self-reported chronic heart disease and symptom-based angina pectoris among middle-aged and older Indian adults. PLoS ONE 18(6): e0287455. https://doi.org/10.1371/journal.pone.0287455

Srivastava, S., Chauhan, S., & Patel, R. (2021). Socio-economic inequalities in the prevalence of poor self-rated health among older adults in India from 2004 to 2014: a decomposition analysis. Ageing International, 46(2), 182-199.

Chauhan, S., Patel, R., & Kumar, S. (2022). Prevalence, factors and inequalities in chronic disease multimorbidity among older adults in India: analysis of cross-sectional data from the nationally representative Longitudinal Aging Study in India (LASI). BMJ open, 12(3), e053953.

4. Please state, how your study is different from the previous existing studies? Justify!

5. Why the authors did not use the Chi-squared test? What do you mean by crude models? Do they measure the same thing – association? Why did you use these in the first place? What was the purpose?

6. As can be seen in Results, the authors have selected some variables that aren't mentioned in the section on explanatory variables. All the explanatory factors should be described in detail. The authors may choose to provide a table describing the various classifications and definitions of each explanatory variable. It increases the authenticity and replicability of work.

7. The mathematical expression of the model used in the study are missing in the methodology section. It is so generic expression. It should include the variables—or at least vectors of variables—for the actual regression in order to increase its value in the paper. As it stands, the generic equation does little beyond the description of variables above.

8. Why did the author included the socio-demographic distribution, if the author has not shown its association with the outcome variables? Justify.

9. What is the major drawback of this study?

10. Tables: All tables should have uniformly 2 decimals.

11. he discussion section is underdeveloped. The authors should highlight how the findings of this study contribute uniquely to the body of existing knowledge .

12. Any additional policy recommendation that focuses on particular states for the specific issues?

13. Please include the source, and captions for each figures properly.

14. There are many grammatical errors. Please revise.

15. Please check the writing format of PLOS. Rearrange the headings and subheadings.

6. PLOS authors have the option to publish the peer review history of their article (what does this mean?). If published, this will include your full peer review and any attached files.

**Do you want your identity to be public for this peer review?** For information about this choice, including consent withdrawal, please see our Privacy Policy.

Reviewer #1: **Yes: **BOTHA, Nkosi Nkosi

Reviewer #2: No

---

## [Decision Letter · Decision Letter 1]

16 Oct 2023

PGPH-D-23-01119R1

Exploring multimorbidity clusters in relation to healthcare use and its impact on self-rated health among older people in India

Dear Dr. Hossain,

Thank you for submitting your manuscript to PLOS Global Public Health. After careful consideration, we feel that it has merit but does not fully meet PLOS Global Public Health’s publication criteria as it currently stands. Therefore, we invite you to submit a revised version of the manuscript that addresses the points raised during the review process.

Please address the remaining concerns by Reviewer#2: Revise your manuscript to provide a clear rationale for your study and contextualise your study relative to previous similar published works.

We look forward to receiving your revised manuscript.

Kind regards,

Miquel Vall-llosera Camps

Staff Editor

Journal Requirements:

2. We have noticed that you have uploaded Supporting Information files, but you have not included a list of legends. Please add a full list of legends for your Supporting Information files after the references list.

Reviewers' comments:

Reviewer's Responses to Questions

**Comments to the Author**

1. If the authors have adequately addressed your comments raised in a previous round of review and you feel that this manuscript is now acceptable for publication, you may indicate that here to bypass the “Comments to the Author” section, enter your conflict of interest statement in the “Confidential to Editor” section, and submit your "Accept" recommendation.

Reviewer #1: All comments have been addressed

Reviewer #2: (No Response)

2. Does this manuscript meet PLOS Global Public Health’s publication criteria? Is the manuscript technically sound, and do the data support the conclusions? The manuscript must describe methodologically and ethically rigorous research with conclusions that are appropriately drawn based on the data presented.

Reviewer #1: Yes

Reviewer #2: Partly

3. Has the statistical analysis been performed appropriately and rigorously?

Reviewer #1: Yes

Reviewer #2: Yes

4. Have the authors made all data underlying the findings in their manuscript fully available (please refer to the Data Availability Statement at the start of the manuscript PDF file)?

Reviewer #1: Yes

Reviewer #2: Yes

5. Is the manuscript presented in an intelligible fashion and written in standard English?

Reviewer #1: Yes

Reviewer #2: No

6. Review Comments to the Author

Reviewer #1: The authors have adequately addressed all issues raised making the manuscript good enough for publication.

Reviewer #2: The comments are not fully addressed. The literature section still required substantial revision. Also, the research question is unclear. The paper is very similar to the previous existing studies. There is no new significant findings.

7. PLOS authors have the option to publish the peer review history of their article (what does this mean?). If published, this will include your full peer review and any attached files.

**Do you want your identity to be public for this peer review?** For information about this choice, including consent withdrawal, please see our Privacy Policy.

Reviewer #1: **Yes: **BOTHA, Nkosi Nkosi

Reviewer #2: No

---

## [Decision Letter · Decision Letter 2]

22 Nov 2023

Exploring multimorbidity clusters in relation to healthcare use and its impact on self-rated health among older people in India

PGPH-D-23-01119R2

Dear Mr. Hossain,

We are pleased to inform you that your manuscript 'Exploring multimorbidity clusters in relation to healthcare use and its impact on self-rated health among older people in India' has been provisionally accepted for publication in PLOS Global Public Health.

Best regards,

Julia Robinson

Executive Editor

Reviewer Comments (if any, and for reference):

Reviewer's Responses to Questions

**Comments to the Author**

1. If the authors have adequately addressed your comments raised in a previous round of review and you feel that this manuscript is now acceptable for publication, you may indicate that here to bypass the “Comments to the Author” section, enter your conflict of interest statement in the “Confidential to Editor” section, and submit your "Accept" recommendation.

Reviewer #1: All comments have been addressed

Reviewer #2: All comments have been addressed

2. Does this manuscript meet PLOS Global Public Health’s publication criteria? Is the manuscript technically sound, and do the data support the conclusions? The manuscript must describe methodologically and ethically rigorous research with conclusions that are appropriately drawn based on the data presented.

Reviewer #1: Yes

Reviewer #2: Yes

3. Has the statistical analysis been performed appropriately and rigorously?

Reviewer #1: Yes

Reviewer #2: Yes

4. Have the authors made all data underlying the findings in their manuscript fully available (please refer to the Data Availability Statement at the start of the manuscript PDF file)?

Reviewer #1: Yes

Reviewer #2: Yes

5. Is the manuscript presented in an intelligible fashion and written in standard English?

Reviewer #1: Yes

Reviewer #2: Yes

6. Review Comments to the Author

Reviewer #1: The authors have adequately addressed all queries raised. Great job done.

Reviewer #2: No further comments.

7. PLOS authors have the option to publish the peer review history of their article (what does this mean?). If published, this will include your full peer review and any attached files.

**Do you want your identity to be public for this peer review?** For information about this choice, including consent withdrawal, please see our Privacy Policy.

Reviewer #1: **Yes: **BOTHA, Nkosi Nkosi

Reviewer #2: No
